# Transcriptome Analysis of Beet Webworm Shows That Histone Deacetylase May Affect Diapause by Regulating Juvenile Hormone

**DOI:** 10.3390/insects13090835

**Published:** 2022-09-14

**Authors:** Jin Cui, Kejian Lin, Linbo Xu, Fangzheng Yue, Liangbin Yu, Quanyi Zhang

**Affiliations:** 1Institute of Grassland Research, Chinese Academy of Agricultural Sciences, Hohhot 010010, China; 2Center for Biological Disaster Prevention and Control, Chinese National Forestry and Grassland Administration, Shenyang 110034, China

**Keywords:** *Loxostege sticticalis* L., diapause, weighted gene co-expression network analysis module, histone deacetylase

## Abstract

**Simple Summary:**

Diapause is a seasonal adaptation to stress in insects, including in Lepidoptera. Juvenile hormone (JH) is the key factor affecting larval diapause. The diapause of beet webworm, an important agricultural pest, is induced by photoperiod, but the mechanism is unknown. Transcriptome sequencing was performed in five different stages of beet webworm. The results showed that 393 genes in the red module were strongly related to JH, and the hub gene of the red module was histone deacetylase (HDAC). After injecting HDAC chemical inhibitors into beet webworms, we found that HDAC enzyme had an effect on JH content in beet webworms under photoperiod that induced diapause. Therefore, HDAC may be involved in diapause in beet webworms by altering JH levels.

**Abstract:**

The beet webworm (*Loxostege sticticalis* L.) is an important agricultural pest and can tolerate harsh environmental conditions by entering diapause. The diapause mechanism of beet webworm is unknown. Therefore, we conducted a transcriptomic study of the process from diapause induction to diapause release in beet webworms. The results revealed 393 gene modules closely related to the diapause of beet webworm. The hub gene of the red module was the HDACI gene, which acts through histone deacetylase (HDAC) enzymes. HDAC enzyme activity was regulated by the light duration and influenced the JH content under induced beet webworm diapause conditions (12 h light:12 h dark). In addition, transcriptomic data suggested that circadian genes may not be the key genes responsible for beet webworm diapause. However, we showed that the photoperiod affects HDAC enzyme activity, and HDAC can regulate the involvement of JH in beet webworm diapause. This study provided a new module for studying insect diapause and links histone acetylation and diapause at the transcriptome level.

## 1. Introduction

The beet webworm, *Loxostege sticticalis* L. (Lepidoptera: Pyralidae), is a cosmopolitan insect that feeds on a wide variety of crop and weed species [1]. Beet webworm is generally considered as a dangerous agricultural pest, similar to the migratory locust *Locusta migratoria* L. [2]. The cyclical phenomenon of beet webworm outbreaks has attracted the attention of researchers [3,4,5]. However, the overwintering pattern of beet webworm and the mechanisms involved have not been widely examined. Beet webworm overwinters through diapause; the larvae enter diapause during short daylight hours in autumn. Under laboratory conditions, the fifth instar of the beet webworm is the photosensitive stage of diapause, which occurs after a short light in the fifth instar stage. However, the mechanism of diapause remains unknown. For example, larvae enter diapause in autumn under short sunlight conditions, do not receive energy intake for up to 6 months, can survive at −20 °C in winter, and can release diapause the following spring, how larvae receive light signals to enter diapause, the regulation of gene expression during diapause maintenance, how larvae resist low-temperature stress at −20 °C during diapause, and the mechanism of diapause release is unclear. Thus, the population dynamics of the beet webworm are not well-understood, and predicting beet webworm outbreak remains challenging.

In insects, the diapause of growth and development occurs after adverse environmental signals, which is a seasonal adaptation strategy for insects to avoid unfavorable conditions [6]. Diapause can occur at any stage of insect growth and development but generally occurs at a fixed growth stage in an insect species. The different stages of diapause can be divided into egg diapause, larval diapause, pupal diapause, and adult diapause [6]. The beet webworm enters diapause in the larval stage [7]. Insect diapause is a complex behavior regulated by environmental and endocrine factors. Insects first sense specific environmental signals that are transmitted to the central nervous system. The response signals act on the endocrine system and determine whether insects should enter diapause through various hormones. Insects prepare for diapause such as by increasing their energy reserves via the joint regulation of the endocrine system and nervous systems [8]. Similarly, insects undergoing the slow development process of diapause are gradually released from this state through the regulation of environmental signals and endocrine systems and eventually return to normal growth and development [8]. Although it is generally accepted that the endocrine regulation of insects induced by environmental signals is the main mechanism directing insect diapause, the details of this process are unclear [9,10,11,12].

Insect diapause is a complex response to environmental changes and depends on changes within the nervous and endocrine systems. The larval diapause that occurs in the beet webworm is common in Lepidoptera, particularly in last instar larvae; there are also reports of first instar larval diapausing [13]. Under larval induction conditions, high concentrations of juvenile hormone (JH) inhibit the release of prothoracicotropic hormone (PTTH), leading to a blockage of ecdysis hormone synthesis, which inhibits larval metamorphosis and stalls insect development at the larval stage. The role of JH in maintaining larval morphology and traits is accomplished in conjunction with ecdysis hormone € in a continuous manner. In Lepidoptera, in the presence of JH, E secretion can only cause larvae to molt and prevent them from pupating. In the final larval stage, when JH levels in the hemolymph gradually decrease and disappear, E secretion can cause molting and pupation. Hiruma et al. [14] showed that larvae contain larval and pupa-specific epidermal protein genes. The synthesis of larval epidermal proteins in the larval molt is temporarily inhibited by the action of E, and after molting, in the presence of JH, the synthesis of larval proteins is resumed, and the larval form is maintained. When the insect continues to pupate at the end of the larval stage, the JH content decreases dramatically and the synthesis of larval epidermal proteins stops completely because of the presence of E, whereas the synthesis of pupal epidermal proteins is promoted [14]. Therefore, the upregulation of JH synthesis during the photosensitive stage is the key to larval diapause [13,15].

Insect diapause is determined by changes in genetic levels [8]. Environmental signals are key to inducing polyphenism in insects [16], and this developmental plasticity is closely related to epigenetic inheritance [17]. Histone acetylation is a kind of epigenetic inheritance, and a large number of transcriptomic studies have shown that it is related to insect diapause [18,19,20]. Diapause in *Allonemobius socius* (Orthoptera: Gryllidae) is maternally regulated, and early embryos at 4–6 days are induced to enter diapause by short-day treatment [21]. The expression of reptin (responsible for encoding the Tip60 histone acetylation complex) is upregulated two-fold in embryos before entering stasis [19]. The short-day treatment of fourth instar larvae or pupae of *Culex pipiens* (Diptera: Culicidae) causes diapause in adult females [22]. Transcriptome profiling revealed a significant overexpression of genes involved in chromatin remodeling in *Culex pipiens* pupae reared under short daylight conditions, including some genes encoding components of the NuRD and Sin3 chromatin remodeling complexes [20]. NuRD is an ATP-dependent multiprotein complex that promotes chromatin cohesion and represses gene transcription [23]; the Sin3 complex, which contains HDAC1 and HDAC2, represses transcription and is associated with stall-related biological processes, including metabolism [24], cell cycle [25], stress tolerance and longevity [26]. *Sarcophaga bullata* (Diptera: Sarcophagidae) enters diapause at the pupal stage after short daylight treatment at the egg or first instar larval stage [27], accompanied by a 75% reduction in the level of histone H3 acetylation modification, a significant reduction in deacetylase activity and a decrease in the expression of the associated coding genes HDAC3, HDAC6, Sirt1, and Sirt2 [18]. Histone deacetylation is typically associated with gene silencing, so reduced levels of histone H3 acetylation may lead to the downregulation of gene transcription, consistent with the characteristics of stunt breeding [8]. The key factor that induces diapause in beet webworms is well-known, i.e., light conditions [28]. Although studies related to diapause have started for a long time [13,29,30], the mechanism of diapause is still not fully understood. In the present study, the light stimulus was used to induce beet webworm into diapause [28,31], and JH was found to be the most critical endocrine regulator affecting beet webworm diapause [13,29,30]. This study improves the understanding of the transcriptional basis of diapause and provides theoretical support for studies of the diapause mechanism in beet webworms.

## 2. Materials and Methods

### 2.1. Insects

The diapause beet webworm was obtained from pre-rearing and purification in our laboratory and was originally from the National Perennial Forage Germplasm Resource Library of the Institute of Grassland Research, Chinese Academy of Agricultural Sciences (Beijing, China). This purified strain was obtained after several generations of successive breeding. Diapause in beet webworms was induced by exposure to a light (L): dark (D) cycle of 12 h each at 21 °C.

More than 1000 adult beet webworms were transported to the laboratory in May 2021 at the experimental base of the Institute of Grassland Research, Chinese Academy of Agricultural Sciences, and reared on natural leaves of *Chenopodium album* L. for three generations before being used in the experiment. The diapause induction training of the beet webworms was performed as described previously [4,28]. The beet webworms were induced to diapause at 21 °C under light conditions of L:D = 16:8 from the first to the fourth instar, and at 21 °C under light conditions of L:D = 12:12 after the end of the fourth instar (NBS). The feeding conditions of the control group and the experimental group were identical, but the control group was fed under 16L:8D light conditions after entering the fifth instar. After the beet webworms entered the soil, we cut down one end of the beet webworm cocoons to observe development. The control group was pupated one week after entering the soil. The control group pupated for more than one week, and the larvae of the diapause treatment group remained in the larval state, and the larvae of the diapause treatment group were named as diapause maintenance 1 (D_Mail1). The larvae after two weeks of diapause were named as late diapause larvae (diapause maintenance 2) (D_Mail2), and after two weeks of diapause, they were placed in the refrigerator in the dark at 4 °C for 3 weeks, and then treated as cold treatment larvae (Chil). One week after low-temperature treatment, we observed pupated larvae from one end of the cocoon of the same batch of beet webworms, indicating that the beet webworms were diapause released, thus named as the sample of diapause is released (Rel_D). All beet webworm species were kept in 28 × 18 × 13.5 cm boxes, 50 per box.

Transcriptome sequencing was performed for five insect states, NBS, D_Mail1, D_Mail2, Chil, and Rel_D, and related physiological and biochemical contents and gene expression profiles were analyzed (Figure 1). Twenty samples were collected from each treatment sample and repeated three times, with each treatment sample collected at an interval of 15 days. Additionally, each sample was collected near the end of the light period.

### 2.2. Transcriptome Sequencing and Bioinformatics Analysis

#### 2.2.1. Nucleic Acid Extraction

Total RNA was extracted from liquid nitrogen-treated beet webworm bodies using TRIzol^®^ Reagent (Plant RNA Purification Reagent, Carlsbad, CA, USA) according to the manufacturer’s instructions (Invitrogen, Carlsbad, CA, USA), and genomic DNA was removed using DNase I (TaKara, Otsu, Japan). The integrity and purity of the total RNA were determined using a 2100 Bioanalyzer (Agilent Technologies, Santa Clara, CA, USA) and quantified using the ND-2000 (NanoDrop Thermo Scientific, Waltham, MA, USA). Only high-quality RNA samples (OD260/280 = 1.8–2.2, OD260/230 ≥ 2.0, RNA integrity number ≥8.0, 28S:18S ≥1.0, >1 μg) were used to construct sequencing libraries.

#### 2.2.2. Library Preparation, Illumina Sequencing, De Novo Assembly and Annotation

RNA purification, reverse transcription, library construction, and sequencing were performed at Shanghai Majorbio Bio-pharm Biotechnology Co., Ltd. (Shanghai, China) according to the manufacturer’s instructions (Illumina, San Diego, CA, USA). The beet webworm RNA-seq transcriptome libraries were prepared using an Illumina TruSeqTM RNA sample preparation Kit (San Diego, CA, USA). Poly(A) mRNA was purified from total RNA using oligo-dT-attached magnetic beads and then fragmented in fragmentation buffer. Using these short fragments as templates, double-stranded cDNA was synthesized using a SuperScript double-stranded cDNA synthesis kit (Invitrogen, Carlsbad, CA, USA) with random hexamer primers (Illumina, San Diego, CA, USA). The synthesized cDNA was subjected to end-repair, phosphorylation, and ‘A’ base addition according to Illumina’s library construction protocol. The libraries were size selected for cDNA target fragments of 200–300 bp on 2% Low Range Ultra Agarose, followed by PCR amplification using Phusion DNA polymerase (New England Biolabs, Boston, MA, USA) for 15 PCR cycles. After quantification by TBS380, two RNAseq libraries were sequenced in single lane on an Illumina Hiseqxten/NovaSeq 6000 sequencer (Illumina, San Diego, CA, USA) to generate 2 × 150 bp paired-end reads. The raw paired-end reads were trimmed and quality controlled using SeqPrep (https://github.com/jstjohn/SeqPrep, accessed on 20 September 2021) and Sickle (https://github.com/najoshi/sickle, accessed on 20 September 2021) with default parameters. A new assembly was generated using Trinity (http://trinityrnaseq.sourceforge.net/, accessed on 20 September 2021). The Cluster of Orthologous Groups, Gene Ontology (GO), and Kyoto Encyclopedia of Genes and Genomes (KEGG) databases were used to determine the highest sequence similarity to a given transcript to retrieve its function. The sequences were annotated, and a typical cutoff E value of less than 1.0 × 10^−5^ was set.

#### 2.2.3. Differential Expression Analysis and Functional Enrichment

RSEM (http://deweylab.biostat.wisc.edu/rsem/) (accessed on 20 September 2021) [32] was used to quantify gene abundances. Differential expression analysis was performed using DESeq2 [33]/EGseq [34]/EdgeR [35] with a Q value ≤ 0.05, differentially expressed genes (DEGs) with |log2FC| > 1 and Q value ≤ 0.05 (DESeq2 or EdgeR) or Q value ≤ 0.001 (DEGseq) were considered as significant. In addition, functional enrichment analyses including GO and KEGG were performed to determine which DEGs were significantly enriched, and GO functional enrichment and KEGG pathway analyses were conducted using Goatools (https://github.com/tanghaibao/Goatools accessed on 20 September 2021) and KOBAS (http://kobas.cbi.pku.edu.cn/home.do accessed on 20 September 2021).

#### 2.2.4. Time Series Analysis

Time series analysis can analyze dynamic gene expression and clusters (genes with a similar expression trend in the five stages were divided into a profile) [36]. We used STEM software to perform this analysis for basic clustering of differentially expressed genes (DEGs) and selected the Clusters with high correlations. Because of the large number of DEGs, we chose 50 temporal patterns with a significant *p*-value of 0.05 using the sequential clustering algorithm: SCM, and maximum time interval: 1.

#### 2.2.5. Identification of DEGSs and Co-Expression Network Module

DEGs were identified at each of the five development stages. The significant DEGs were screened as described in Section 2.5 and used as basic data for this analysis. Only DEGs with mean expression values greater than 1 for the three samples at each stage were retained, and the coefficient of variation was 0.05 [37,38]. The highly co-expressed gene modules were inferred from the DEGs using weighted gene co-expression network analysis (WGCNA) with the parameter settings of Merge Cut Height: 0.25. Min KME to Stay: 0.3, min Module Size: 30, Soft Power: 5, coefficient of variation: 0.05.

#### 2.2.6. Gene Set Enrichment Analysis

As the only environmental conditions that changed were temperature and light, we performed a Gene set enrichment analysis (GSEA) of biological clock genes and thermogenic genes. GSEA can reveal whether certain genes are enriched in the circadian gene/thermogenic gene set and whether the genes in the gene set are randomly distributed or enriched at the top or bottom of the corresponding gene set. The significant enrichment of genes at the top or bottom of the set indicates that the expression of these genes significantly affects differences in pre-grouping [39].

We used RSEM software with default parameters for this analysis. Beet webworms in the five stages were divided into 10 groups by performing GSEA for each two.

### 2.3. Quantitative Reverse Transcription RT-PCR Analysis

GAPDH was used as a reference gene for quantitative RT-PCR. Premier 5.0 software (Premier Biosoft, Palo Alto, CA, USA) was used to design the primers, and 80–200 bp products were selected (Appendix A). Three biological replicates and three technical replicates were evaluated for each gene q-PCR system.

DNase-treated RNA (1 mg) was reverse transcribed in a 20 µL reaction. Reverse transcription was performed in a 20 µL reaction mixture using a cDNA Reverse Transcription Kit (Novoprotein Scientific, Summit, NJ, USA). Gene-specific primers were designed using Premier 5.0 software (Appendix A). Quantitative reverse transcription PCR (qRT-PCR) assays were performed using 20 ng cDNA, 2 µL primers, 0.4 µL ROX I, RNase-free water to 10 µL, and SYBR Green I qPCR SuperMix Plus (Novoprotein Scientific, Summit, NJ, USA) to 10 µL. Each sample was evaluated in three biological replicates and three technical replicates on an ABI STEP ONE Plus (Applied Biosystems, Foster City, CA, USA) with initial denaturation at 95 °C for 1 min, followed by 40 cycles of 95 °C for 20 s and 60 °C for 1 min.

The relative cycle threshold (CT) of each was averaged across the technical replicates. The resulting CT value of each gene was normalized to the geometric mean of the CT value of the reference gene (GAPDH) by subtracting the mean CT of the reference gene from the CT value of the gene (2−ΔCT method).

### 2.4. Measurement of JH content and HDAC Enzyme Activity

JH1 was determined using an enzyme-linked immunosorbent assay (ELISA) kit (Shanghai Hengyuan Biotechnology Co., Ltd., Shanghai, China). The detection level of the JH kit is 6.25–250 pg/mL, and the detection sensitivity is less than 1.0 pg/mL. HDAC activity was assayed using ELISA kits (Shanghai Tongwei Industrial Co., Ltd., Shanghai, China). The assay range of the kit is 4–220 U/L. Three biological replicates and three technical replicates were evaluated in both the JH1 and HDAC assays.

### 2.5. Injection of HDAC Inhibitor into Beet Webworm

An HDAC inhibitor (Beijing Solarbio Science and Technology, Beijing, China) was diluted to 1:100 and injected using a microinjector into the beet webworms, followed by culture under light conditions of induced diapause, i.e., the beet webworm cultured at the fifth instar under 12L:12D conditions for 24 h. The injection site was between the 4th ventral node of the tail of the beet webworm, and 45 μL was injected per beet webworm larvae, with 30 injections per treatment. The inhibitor control was water, and the light was controlled at 16L:8D. The HDAC enzyme activity and the JH content were measured at 48 h after injection.

## 3. Results

### 3.1. Simple Analysis of Transcriptome Data

A total of 220 GB of raw data were obtained using the double-end sequencing method. A total of 106,409 genes were assembled. All subsequent analyses were based on these genes. The validation of the beet webworm transcriptome data showed that the gene expression trends were consistent with the transcriptome data, indicating that the transcriptome data were authentic and reliable (Figure 2).

#### 3.1.1. Identification of DEGs

For the five stages, DEGs were identified as those showing ≥2 fold up- or downregulated transcriptome expression, and transcriptome comparison was performed every two stages to identify the DEGs. A total of 11,486 non-redundant DEGs were identified, which included NBS/D_Mai1, 3144; NBS/D_Mai2, 7064; NBS/Chil, 4402; NBS/Rel_D, 4592; D_Mai1/D_Mai2 375; D_Mai1/Chil, 1034; D_Mai1/Rel_D 990; D_Mai2/Chil, 2024; D_Mai2/Rel_D, 1286; and Chil/Rel_D 598 (Figure 3, Appendix A).

#### 3.1.2. Analysis of Differences in Time Series Expression

The 11,486 non-redundant differential genes were divided into 50 profiles according to their expression trends, Among the 50 profiles (0–49), 15 profiles showed significant trends in 45, 3, 2, 6, 0, 49, 1, 9, 20, 47, 30, 48, 8, 44, and 4 (Appendix A). Additionally, these 15 profiles were divided into eight clusters based on their similar expression trends (Figure 4).

Each cluster was subjected to KEGG enrichment analysis to identify enriched genes. Cluster1 was enriched in 299 KEGG pathways, among which 21 pathways were significant, including 11 metabolism-related pathways (map00290, map00630, map00620, map00513, map00270, map00190, map00020, map00260, map00350, map00380, map00010) and 10 tissue system pathways (map04913, map04213, map04976, map04212, map04714, map04742, map04211, map map04925, map04750, map04260) (Appendix A).

Cluster2 was enriched on 306 KEGG pathways, excluding the human diseases pathway, of which 45 pathways were significant, including 6 cellular processes (map04142, map04530, map04810, map04144, map04145, and map04510), 4 environmental information processing (map04080, map04512, map04514, map04151), 20 metabolism pathways (map00040, map00983, map00053, map map00140, map00600, map00480, map00830, map00982, map00860, map00561, map01040, map00980, map00100, map00240, map00564, map00590, map00920, map map00900, map00300, map00790), and 15 organismal systems pathways (map04974, map04972, map04640, map04730, map03320, map04666, map04672, map04614, map map04979, map04970, map04624, map04966, map04360, map04975, map04658) (Appendix A).

Cluster3, Cluster5, and Cluster7 were enriched in fewer pathways of significance. Cluster3 had five pathways, including one cellular processes pathway (map04142), three metabolism pathways (map00100, map00983, map00590), and one organismal systems pathway (map04624) (Appendix A). Cluster5 has three cellular processes on (map04214 (nine genes), map04210 (10 genes), map04215 (five genes)), one environmental information processing (map04151), and one genetic information processing pathway (map00970 (seven genes) (Appendix A). Cluster7 has nine entries, one cellular process (map04145), three genetic information processing (map04141, map03060, map00970), one metabolism (map00510), and four organismal systems pathways (map04966, map04721, map04918, and map04361) (Appendix A).

Cluster4, Cluster 6, and Cluster 8 mostly contained pathways with significant enrichment in metabolism. Cluster4 included 29 pathways, with 24 metabolic pathways (map00620, map00340, map00330, map00380, map00010, map00410, map00981, map00310, map00071, map00280, map00561, map00053, map00500, map00230, map00260, map00061, map00250, map00030, map00640, map00770, map00311, map00472, map00520, map00270) (Appendix A). Cluster6 has 19 entries, 16 enriched in metabolic pathways (map00980, map00982, map00983, map00053, map00140, map00830, map00480, map00860, map00280, map00071, map00040, map00100, map00981, map00330, map00410, map00260) (Appendix A). Cluster8 contained 12 metabolic pathways (map00561, map00790, map00051, map00040, map00052, map00981, map00670, map00600) (Appendix A).

Most of the enriched pathways were related to metabolism. Thus, the diapause of beet webworms may be dominated by metabolism.

### 3.2. Identification of WGCNA Modules Associated with JH

The highest JH content was observed before the diapause of beet webworm, i.e., at the diapause induction stage. This content significantly differed from that in all other stages. The JH content increased in beet webworms subjected to low-temperature treatment, but the increase was not significant (Figure 5).

After data preprocessing, 7912 non-redundant genes were analyzed to identify 10 WGCNA modules (Figure 6), and 393 genes with red modules were highly associated with JH (r = 0.851, *p* = 5.76 × 10^−5^), (Figure 7, Appendix A).

According to the results of WGCNA, the red module showed the highest correlation with JH. Therefore, 393 genes in the red module played important roles in changing the JH content in the beet webworms and thus were involved in the diapause of the beet webworms (Figure 7, Appendix A).

The enrichment analysis of these 393 genes using KEGG showed that most of the genes were related to metabolism. In the second category, carbohydrate metabolism, lipid metabolism, endocrine system, and signal transduction accounted for most of the genes (Figure 8).

We chose *Chilo suppressalis*, which belongs to the Lepidoptera borer moth family in the field of entomological taxonomy [40], as the reference species for the red modules and analyzed the interaction networks (Figure 9). The results revealed that three genes in the reciprocal network had larger values of betweenness centrality. Therefore, the nodes of the three genes play the most important role in ensuring that the whole network remains tightly connected. The values of closeness centrality were also higher for these three nodes (Figure 9). However, the highest degree of centrality among these three genes was the gene with sequence number CAB3251393.1 in NCBI and our query results of NAD-dependent HDAC activity and the presence of the KEGG cell cycle pathway in GO. The result of the Cluster of Orthologous Groups query was HDAC, which compresses the chromatin structure and causes it to be difficult for RNA polymerase to enter, thus reducing gene expression.

### 3.3. HDAC Inhibitor Suggests That HDAC Affect JH Content

The HDAC enzyme activity and JH content of the beet webworms were measured 48 h after the injection of the HDAC inhibitor. The HDAC enzyme activity and JH content were significantly higher under 12L:12D photoperiodic conditions compared to under 16L:8D conditions. Under photoperiodic conditions that induced the diapause of the beet webworms, i.e., the 12L:12D conditions, the HDAC enzyme activity of the beet webworms following the HDAC inhibitor injection significantly decreased under 12L:12D conditions and significantly reduced the JH content, and under the 16L:8D condition, the HDAC inhibitor did not significantly alter the JH content of the beet webworms (Figure 10).

### 3.4. GSEA Results Showed a Significant Effect of Thermogenic Genes on Diapause

As only light conditions and temperature varied throughout the process of the *Loxostege sticticalis*, we performed a GSEA of the genes annotated to the phototransduction pathway (map04744, map04745), circadian rhythm pathway (map04710, 04711, 04713) and the thermogenesis pathway (map04714), and the results showed that the biological clock genes did not significantly affect the diapause of *Loxostege sticticalis* (Figure 11A, Appendix A). In contrast, genes annotated to the thermogenesis pathway (map04714) showed significant effects, except for NBS/Rel_D, D_Mail1/D_Mail2 (Figure 11B, Appendix A). From the NBS to the D_Mail1/D_Mail2 stages, there was no change in temperature, but genes in the thermogenesis pathway had a significant effect. D_Mail1 and D_Mail2 were in the diapause maintenance stage when the environmental factors did not change, and genes in the thermogenesis pathway did not significantly affect the difference between D_Mail1 and D_Mail2.

## 4. Discussion

The mechanism of insect diapause has been studied for several decades. These studies showed that entry into diapause is not simply a shutdown in gene expression [8,41]. We performed transcriptome sequencing of the non-diapause period, two stages of the diapause maintenance period, low-temperature treatment period, and diapause release period, and DEGs were screened out for further analysis. In all, 11,486 DEGs were identified in this study, which may include genes most closely associated with the diapause of *Loxostege sticticalis*. These genes also included not only the DEGs in a pre-diapause state, diapause state, and diapause-released state of the *Loxostege sticticalis* but also DEGs following low temperature during diapause. We could not directly select genes from among the 11,486 genes with key roles in the diapause of *Loxostege sticticalis*. Therefore, we used time-series expression analysis data and found eight clusters; functional enrichment analysis of these clusters showed that most of the enriched KEGG pathways belonged to metabolic pathways. Notably, we were unable to obtain specific DEGs by using time series analysis. We only used the function of time series analysis to cluster the expression pattern of the transcriptome.

JH is the most important factor in the diapause of lepidopteran larvae. We detected the JH content of the beet webworm in these five stages and found that, in the pre-diapause stage, the JH content of the beet webworm was significantly different from other stages. In this study, the JH content of the light-sensitive stage of the beet webworm, the fifth instar beet webworm, increased significantly when subjected to photoperiodic conditions that induced diapause, which was the key to the entry of the beet webworm into diapause. The JH content of the beet webworms did not increase significantly after the diapause was lifted. We performed trait–gene analysis of the JH. The WGCNA of the 11,486 DEGs revealed a WGCNA ‘red’ module of 393 genes that were highly correlated (r = −0.851, *p* = 5.76 × 10^5^) with fat JH contents.

We consider 393 genes in the red module, which are most associated with the JH as the key set of genes regulating the diapause of the beet webworm. We analyzed the protein interaction network of the red module and found that the HDAC gene played an important role in the whole network. We examined the enzymatic activity of HDAC and found that the enzymatic activity of HDAC was regulated by light, i.e., there was a significant increase in the enzymatic activity of HDAC under 12L:12D conditions compared to 16L:8D. Additionally, under the light conditions that induced the diapause of the beet webworm, there was a significant decrease in JH content after the use of an HDAC chemical inhibitor. However, this phenomenon was not observed under normal culture conditions, i.e., under 16L:8D conditions. Therefore, we have sufficient evidence that HDAC is involved in the diapause of beet webworm by regulating JH.

We also conducted GSEA analysis on NBS, D_Mail1, D_Mail2, Chil, and Rel_D of beet webworm to test whether circadian clock genes and thermogenic genes were responsible for the changes between any two stages. The results showed that circadian clock genes did not play a significant role in the diapause and diapause release of beet webworm. Thermogenic genes are responsible for the differences between each stage. This may be because the regulatory networks of the circadian clock signals are extremely complex, and the analysis results are not significant [42]. However, the regulation of the thermogenic pathway, which is directly involved in energy metabolism, is more direct than that of clock genes. Additionally, HDAC can regulate histone deacetylation, which can be linked to cellular thermogenesis, and that HDAC can regulate the thermogenic process of lipids [43,44]. In any case, we found that the photoperiod affected the enzymatic activity of the HDAC of the beet webworm and that HDAC was involved in the diapause of beet webworms by affecting the JH content.

Numerous studies showed that histone acetylation modifications are associated with insect diapause [18,20,21]. *Allonemobius socius* showed a significant increase in the expression of REPTIN (responsible for encoding the Tip60 histone acetylation complex) in embryos before entering diapause [21]. Short daylight exposure of the fourth instar larvae causes female adults to enter diapause [22]. Transcriptomic studies revealed a significant overexpression of genes involved in chromatin remodeling processes in *Culex pipiens* nymphs reared under short daylight conditions, including HDAC1 and HDAC2 [20]. Histone deacetylation may prolong the life span of *Culex pipiens* by causing a four-fold increase in the gene encoding Pax protein, a downstream target of FOXO, an important diapause regulator in *Culex pipiens* [45], which is involved in the localization of Sirtuin 1/Sir 2 (a NAD-dependent HDAC) and the silencing of the protein responsible for regulating life extension [46,47]. Histone deacetylation is typically associated with gene silencing; thus, reduced levels of histone acetylation may lead to the downregulation of gene transcription, which is consistent with the characteristics of diapause [8,48] Diapause is maintained through the kat7-mediated acetylation of H4 and Baz2B-related acetylation of H3 in *Tetrapedia diversipes* (Apidae: Tetrapediini). The expression of the histone acetyl transferase (Kat7), histone H3, and bromodomain adjacent to zinc finger domain protein 2B (BAZ2B) genes were significantly upregulated during the diapause stage. The expression of the histone genes H2A and H2B was downregulated [49]. The acetylation of histone H3 can be involved in the induction, maintenance, and release of chrysalis diapause in *Sarcophaga* flies, suggesting that the low levels of the acetylation of histone H3 are key to the regulation of chrysalis diapause in *Sarcophaga* flies [21].

Diapause can occur at every stage of insect development. Both diapause and development regulation requires the accurate regulation of endocrine signals. Acetylation is a systematic regulatory process with biological functions in various developmental stages of insects. Therefore, epigenetic coupled endocrine regulation may regulate diapause. Previous studies focused on histone deacetylation, which induces 20E synthetic gene expression [20] and participates in JH function [50,51]. In *Tribolium castaneum* (Coleoptera: Tenebrionidae), HDAC1 and HDAC3 regulate metamorphosis by affecting the expression of the JH-responsive gene KR-H1 (Krüppel homolog 1) [51]. In silkworm, exogenous 20E can activate the expression of the cyclic adenosine monophosphate-responsive element binding protein (CREB) in Bm12 cell lines, and induce histone H3K27ac, mediated by CBP (CREB-Binding protein) [52]. These studies indicated that acetylation couples insect endocrine signals and further affects diapause in insects.

Research on the effects of acetylation on insect diapause remains in the early stages. In our further studies, we will evaluate the mechanism of the influence of diapause environmental conditions on acetylation, and then determine the molecular mechanism of acetylation coupled with endocrine regulation. Further, to mediate acetylation regulation, we will examine the cell signal transduction and synergistic regulation of final acetylation in insect diapause. Acetylation may be a molecular marker of diapause regulation or developmental determination and is important for analyzing the specific role of the epigenetic code in diapause regulation to understand the insect diapause mechanism on a wider scale.

## Figures and Tables

**Figure 1 insects-13-00835-f001:**
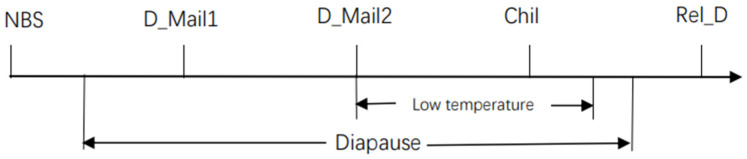
Schematic depiction of experimental design. NBS was sampled on the first day using L:D = 12:12. Both samples were in the diapause stage during diapause and low-temperature treatment at 4 °C. Rel_D is the sample after diapause is released. The exact time of the start and end of diapause is not certain, and the start and end of diapause marked on the picture is an estimate.

**Figure 2 insects-13-00835-f002:**
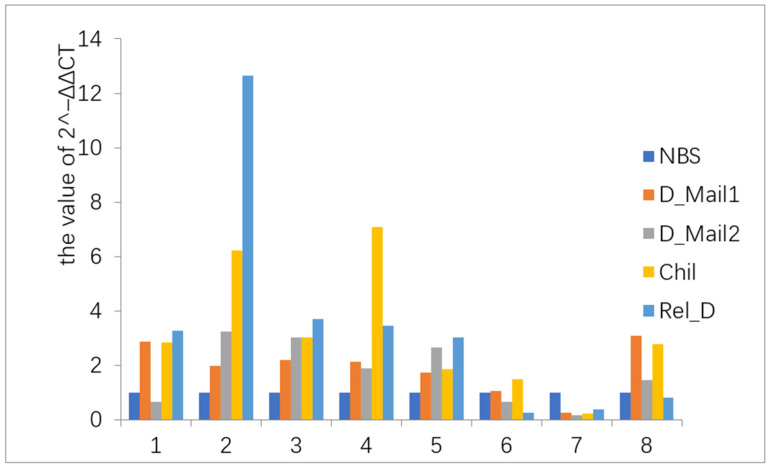
Quantitative reverse transcription RT-PCR analysis. The horizontal coordinates indicate the eight randomly selected genes (Appendix A). The vertical coordinate indicates the value of 2^^−ΔΔCt^. The expression trends of these eight genes in the five phases were the same as the transcriptome, indicating the data are true and reliable. Gene expression levels in the NBS phase were set to 1, whereas gene expression levels in other phases were considered as upregulated for values of more than 1 and downregulated for values of less than 1.

**Figure 3 insects-13-00835-f003:**
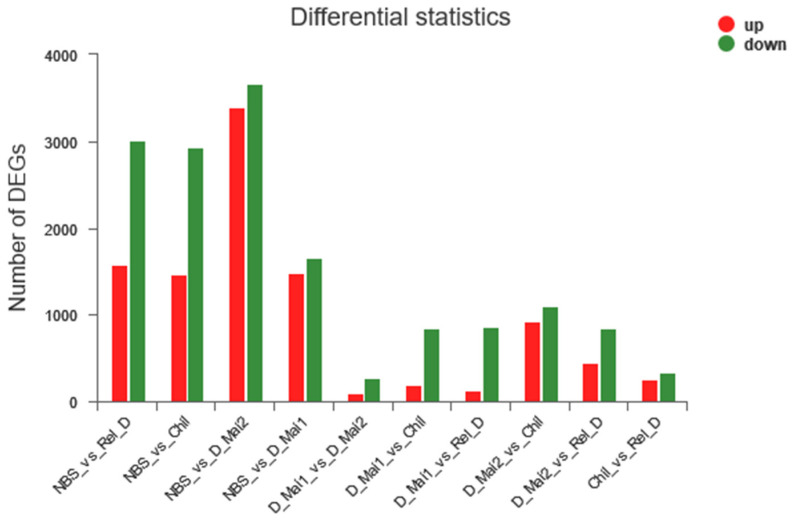
Statistical figure of DEGs at every two stages.

**Figure 4 insects-13-00835-f004:**
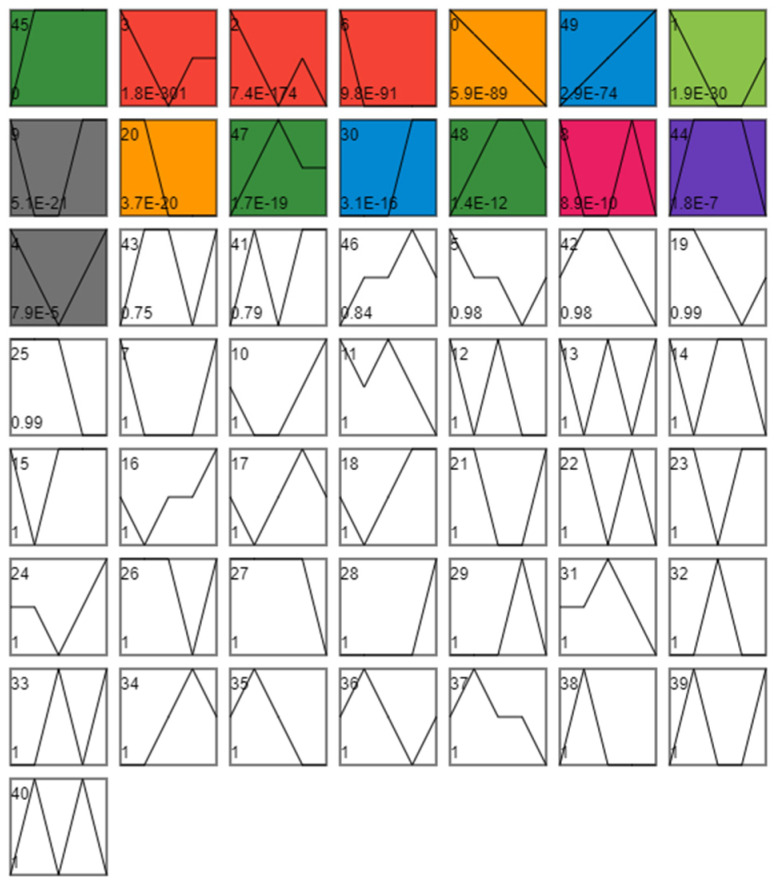
Time series analysis. Each profile corresponds to a rectangle. The number in the upper left corner of the rectangle is the number of the profile, starting from 0, where the broken line is the trend of the expression amount changing over time, and the value in the lower-left corner is its corresponding significance level *p*-value. Trend diagram with color: it indicates that the time-series pattern of the profile conforms to the significant change trend. Profiles with the same color belong to the same cluster (profiles with similar trends belong to the same category—the same color). Figure shows that all DEGs can be divided into 50 profiles according to their expression trends, among which 15 categories have significant gene expression trend changes. Additionally, these 15 profiles with significant expression trend changes were further divided into eight clusters because of similar expression trends (consistent color indicates similar expression trends).

**Figure 5 insects-13-00835-f005:**
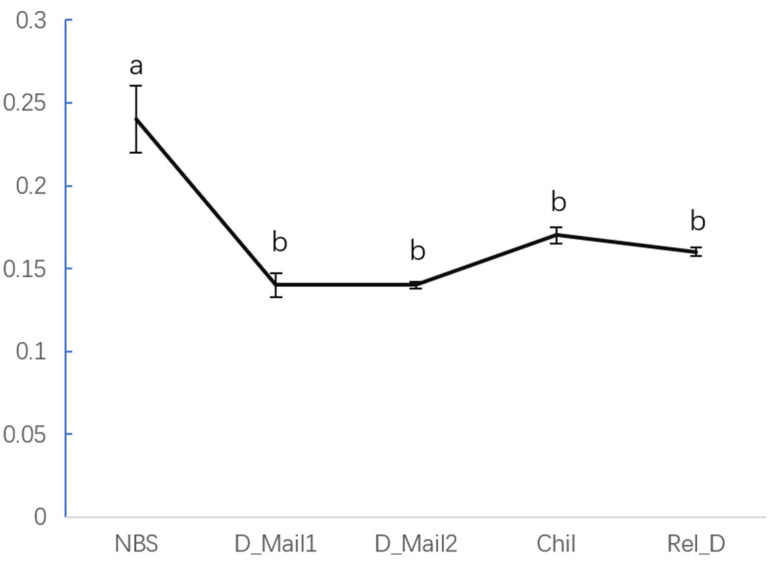
Levels of juvenile hormone 1 (JH1) in various stages of the beet webworm. Different letters represent significant differences.

**Figure 6 insects-13-00835-f006:**
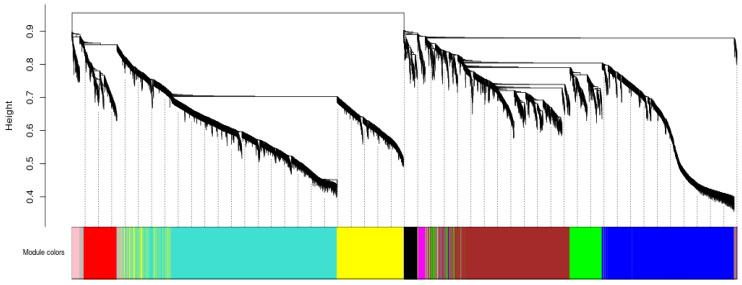
Weighted gene co-expression network analysis (WGCNA). Genes are classified into modules according to their expression trends, where a branch represents a gene, and each color represents a module. A gray color represents a gene that is not classified into a specific module.

**Figure 7 insects-13-00835-f007:**
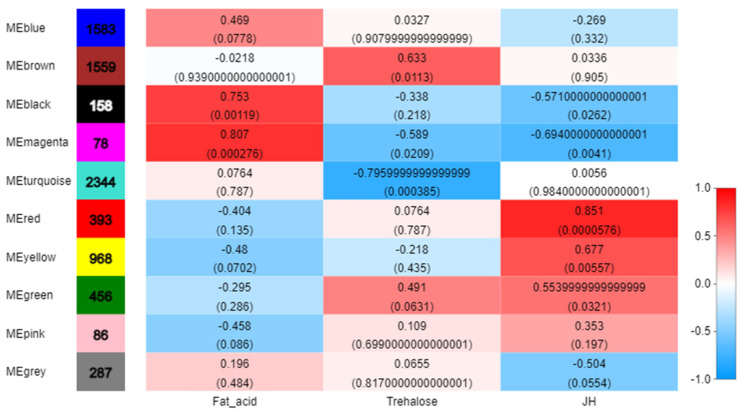
Correlation of modules with each physiological index. Horizontal coordinates represent fatty acids, trehalose, and JH, and vertical coordinates represent different gene modules. The left column of the graph indicates the number of genes in the module, and each set of data on the right indicates the correlation coefficient and significant *p*-value (in parentheses) of the module with the phenotype. The default designation of red and blue means that the module is more and less correlated with the phenotype, as indicated by the numbers under the color bar at the bottom right.

**Figure 8 insects-13-00835-f008:**
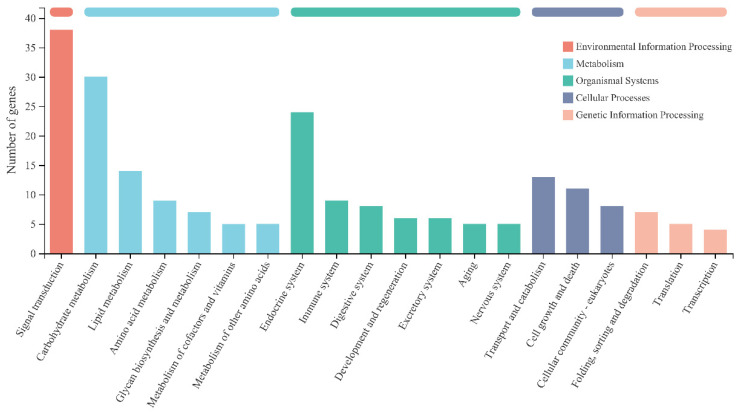
Abscissa shows the KEGG metabolic pathway, and the ordinate shows the number of genes annotated to this pathway.

**Figure 9 insects-13-00835-f009:**
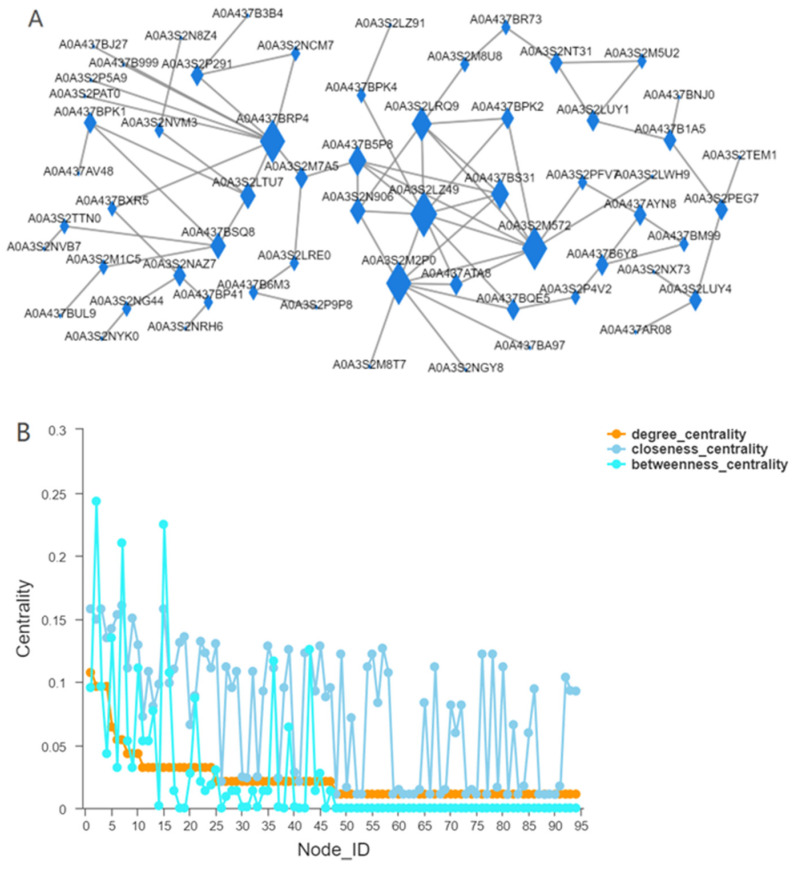
Protein interaction network. (**A**): Nodes represent genes, and edges represent interactions between two genes. The size of a node is directly proportional to the connectivity (degree) of the node; that is, the more edges connected to the node, the larger the degree of the node will be, indicating the stronger the importance of genes in the network. (**B**): The horizontal coordinate is the ID of the node, and the vertical coordinate is Centrality. The larger the value is, the more important the node is in the network. In the figure, the blue color represents betweenness centrality, which reflects the role of a node in connection with other nodes. It means that the node plays a more important role in maintaining the tight connectivity of the whole network); Orange represents degree centrality (degree centrality is the most direct metric to describe node centrality in network analysis. The greater the degree of a node is, the higher the degree centrality of the node is, and the more important the node is in the network). Light blue represents the closeness coefficient (which is the distance between a node and other nodes in the network, if all are very short, the point is the center of the whole, and the larger the value is, the closer the node is to the center of the network).

**Figure 10 insects-13-00835-f010:**
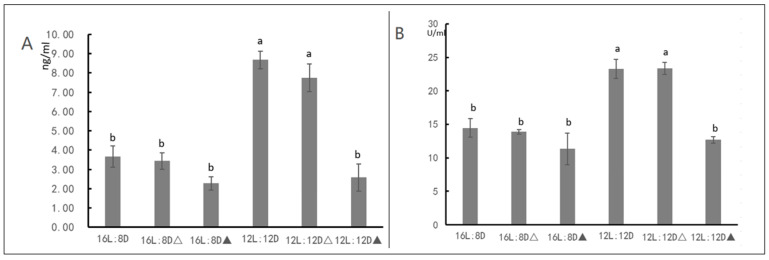
(**A**) HDAC enzyme activity after HDAC inhibitor injection under different light conditions. (**B**) JH content after HDAC inhibitor injection under different light conditions. “△” indicates that an equal amount of distilled water was injected to the beet webworm, “▲” indicates that HDAC inhibitor was injected. Different letters represent significant differences.

**Figure 11 insects-13-00835-f011:**
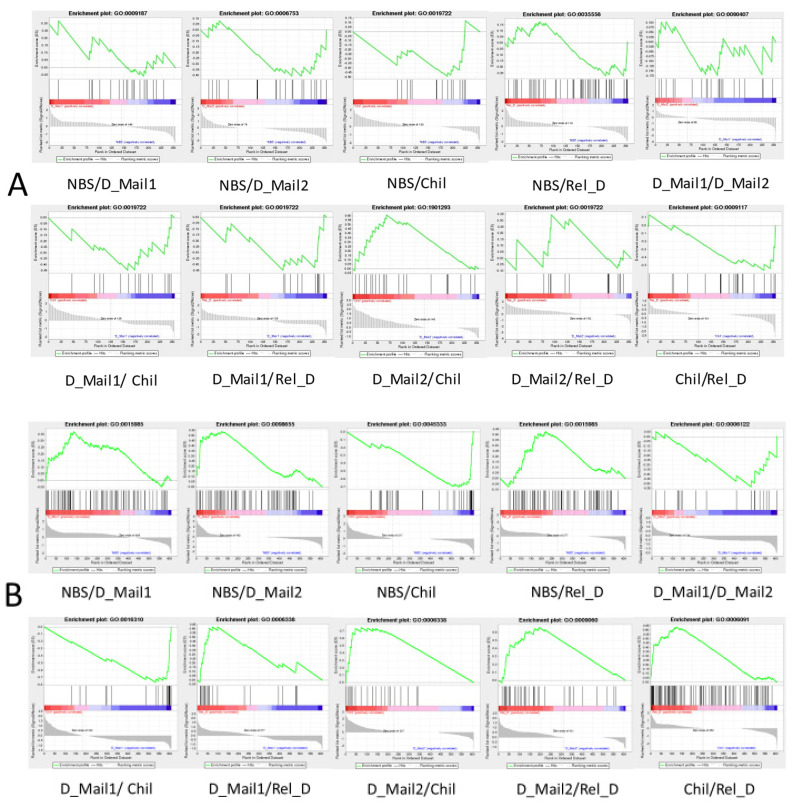
Gene set enrichment analysis (GSEA) (**A**) Phototransduction and Circadian rhythm genes play a role in different periods of beet webworm. (**B**) Thermogenesis gene plays a role in different stages of the beet webworm. The curve at the top represents the dynamic enrichment score (ES) value, and the highest point represents the ES value of the gene set (details of each figure are shown in Appendix A).

## Data Availability

Transcriptome raw data have been uploaded to NCBI, reference: PRJNA837088.

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
