# Peer review of "Transcriptome Analysis of Beet Webworm Shows That Histone Deacetylase May Affect Diapause by Regulating Juvenile Hormone"

_insects, 2022, doi:10.3390/insects13090835_

Round 1

Reviewer 1 Report (Previous Reviewer 2)

Further editing of sections needed to make the paper clear.

Author Response

Thank you very much for taking the time to review this manuscript and your comments on this article. Finally, I have revised this article according to your suggestion, and the paper has been edited by a scientific editor. We look forward to your further comments on this manuscript.

Reference 2 by is not appropriate as it never mentions flying locusts, Locusta migratoria, (there is no locust called Occsta migratoria) but only mentions Schistocerca gregaria, another species of locust.

The first sentence in Reference 2 "The beet webworm Loxostege sticticalis L. shares by right the top position with the migratory locust Locusta migratoria L. in the list of especially dangerous agricultural pests, the webworm outbreaks periodically embracing millions of hectares.” I read this parper and revised Flying locusts Occsta migratoria in the manuscript to migratory locust Locusta migratoria L.

Sentence beginning: “Most of the mechanisms…” should be re-written as punctuation switches between semi-colons and commas, as well as having a capitalised “The” in the middle.

This has been revised in the new manuscript as you suggested.

Sentence beginning “Insect diapause is a complex response in response to changes in the environment and dependent upon changes within the nervous and endocrine systems”First sentence of third paragraph should be re-written.

This has been revised in the new manuscript as you suggested.

What are “molecular levels”?

In biology, the structure and function of organisms are studied from a microscopic (i.e. molecular) perspective, and the mechanisms of their life activities are explored. Research at the molecular level includes studies at the level of genes, proteins, etc. However, I have reviewed a number of papers and did not find this expression “at the molecular level”. Therefore, I changed “molecular level” to “genetic level”.

Remove et al. after pseudogene as that is confusing.

This has been revised in the new manuscript as you suggested.

Need to identify taxa when introducing insects, such as Allonemobius socius, as well as define what “stagger” is.

The "stagger" means diapause, and to avoid misunderstanding, The revised manuscript changed "stagger" entirely to "diapause". The insect taxa has also been added.

Methods

How is dynamic gene expression clustered using time series analysis? Or is the clustering of

patterns of gene expression? Need to re-write to remove “them” at the end of the sentence as that is unclear.

Because the sampling time interval of the five stages is fixed, time series analysis can be used to cluster insects according to the changes in the expression of each gene in the five stages. Genes with similar expression trends were clustered into a profile.

This sentence has been revised and explained in parentheses in the new manuscript.

Using abbreviation DEGs before indicating that stands for differentially expressed genes, must be included with heading

This has been revised in the new manuscript as you suggested.

2.2.3.

Sentence beginning “Three biological replicates…” is not a complete sentence.

This has been revised in the new manuscript.

Need to give source of ELISA kits used for JH and HDAC activity.

The source of the kit is provided in the new manuscript.

Results

Line 298. Change “this” to “these”

This has been revised in the new manuscript as you suggested.

Section 3.2.2

Need to check the punctuation within this section as well as capitalising the beginning

of sentences. The section is confusing without ensuring these components are correct.

This has been revised in the new manuscript as you suggested.

Figure 8. Some of the p values don’t seem to be significant (p<0.05). If not significant, better to present as NS.

We tried to use NS to replace the insignificant P value, but the pictures did not look uniform. After discussion with the co-authors, we still used the value.

Lines 411-440 confuse between results and discussion points. Remove “4. Discussion” from legend to table 11.

This has been revised in the new manuscript as you suggested.

How do you determine what genes of caterpillars are involved in “thermogenesis”?

Genes annotated in the Thermogenesis Pathway (MAP04714) were defined as being involved with "Thermogenesis" as a gene set.

Discussion

Lines 546-565 has several lines that need to be removed as they are redundant.

This has been revised in the new manuscript as you suggested.

Need taxa for Tetrapedia diversipes.

This has been revised in the new manuscript as you suggested.

Reviewer 2 Report (New Reviewer)

By analyzing RNA-seq data from 5 different samples (5 different stages) of beet webworm during diapause inductionmaintenance and termination, the author concluded that histone deacetylase (HADC) affects diapause by regulating juvenile hormone (JH). It is an interesting topic but the writing style of whole manuscript is not logical and must be reorganized!  

Simple summary should not be a copy of abstract!

The subsections of the results part seem no link, better to combine them as a whole story

The authors should carefully check spellings in the whole manuscript such as in the line 43 “Occsta migratoria L.”

English of the manuscript must be revised by a scientific editor or native speaker

The quality of figures must be improved! For examples, Figure 5, 9, 10, 11 and 12

Other major concerns:

Using HDAC inhibitor to treat the short day induced (prediapause) insects, as the authors showed, the concentration of the JH was reduced, I just wonder whether the HDAC inhibitor-treated insects stop entering diapause?

In Figure 2, what about the JH content in the beet webworms reared under long day condition?

In Figure3, which genes selected for validating the RNA-seq data?

The density (number) of the insects used for diapause experiment must be indicated in the materials and methods!

Minor points

Line 165, the insects were placed in refrigerator at 4 °C for 3 weeks, which kept in dark or under light?

Line 182, 2.2.1 Nucleic acid extraction and RNA-seq library construction

No information of the library construction was provided in this section

Reference 4 and 41 are the same!

Author Response

Thank you very much for taking the time to review this manuscript and your comments on this article. Finally, I have revised this article according to your suggestion, and the paper has been edited by a scientific editor. We look forward to your further comments on this manuscript.

By analyzing RNA-seq data from 5 different samples (5 different stages) of beet webworm during diapause induction,maintenance and termination, the author concluded that histone deacetylase (HADC) affects diapause by regulating juvenile hormone (JH). It is an interesting topic but the writing style of whole manuscript is not logical and must be reorganized! 

We reorganized and edited the manuscript. The revised manuscript is structured as follows:

Introduction:

Paragraph 1: Introduce Beet webworm and explain that the diapause mechanism of beet webworm is blank

Paragraph 2: Introduction to diapause in insects, induced by the environment.

Paragraph 3: Diapause in insects is mediated by both the environment and the endocrine system

Paragraph 4: The connection between diapause and acetylation in insects

Paragraph 5:The main environmental factor in this study is light, and explain the significance of the study

Materials and Methods are not structurally adjusted.

The Results section is reorganized as 3.1. Simple analysis of transcriptome data. 3.2. DEGs in 3.1 results were used as the basis for this part, and combined gene phenotype analysis was performed: RED module was the module with the highest correlation with JH, and hub gene of RED module was HDAC. 3.3. HDAC chemical inhibitors were used to prove that HDAC affected JH content. 3.4. Transcriptome analysis was supplemented to detect the role of circadian rhythm genes and thermogenic genes in diapause of beet webworm.

Some adjustments have been made in the Discussion section. The adjustment is based on the Results.

section.1. Simple analysis of transcriptome dataSimple summary should not be a copy of abstract!

The simple summary section has been revised in the manuscript.

The subsections of the results part seem no link, better to combine them as a whole story

We reorganized the results part of the manuscript:

3.1 Simple analysis of transcriptome data to obtain differential genes and perform simple clustering of differential genes.

3.2 WGCNA analysis and combined trait(JH) gene analysis showed that HDAC gene was involved in the change of JH.

3.3 It was proved that different photoperiod affected the enzyme activity of HDAC and HDAC regulated the change of JH.

3.4 It is a supplement to our hypothesis, and the results of the data analysis showed that biorhythm genes were not mainly responsible for diapause in the beet webworm.

Thus, we concluded that HDAC was involved in diapause of beet webwormthrough JH.

The authors should carefully check spellings in the whole manuscript such as in the line 43 “Occsta migratoria L.”

In the new manuscript, "Occsta migratoria L." has been changed to "Locusta Migratoria L."

English of the manuscript must be revised by a scientific editor or native speaker

The revised manuscript has been revised in a professional editorial.

The quality of figures must be improved! For examples, Figure 5, 9, 10, 11 and 12

The quality of the pictures in the article is optimized, and the original pictures are sent to the editorial department when they have been submitted. The quality of the images in the manuscript will improve later.

Other major concerns:

Using HDAC inhibitor to treat the short day induced (prediapause) insects, as the authors showed, the concentration of the JH was reduced, I just wonder whether the HDAC inhibitor-treated insects stop entering diapause?

When after the injection of HDAC inhibitors, we observed the phenomenon of the cultivation of beet webworm is a short light conditions of JH content has decreased, but in the end we don't have to observe them stop diapause, but after the injections, decrease food intake, is in a state of stagnation development, at the time of about 10 days, after the injection of distilled water and HDAC inhibitors of beet webworm all died.

In Figure 2, what about the JH content in the beet webworms reared under long day condition?

As explained in Figure 11B, the JH content of the beet webworm under long day condition was about 40% of that under short day condition.

In Figure3, which genes selected for validating the RNA-seq data?

The eight genes are:

1

2

3

4

5

6

7

8

CRY

sp

ALDH

VIR

Per2

TIM

cyc

PDP

The names of these genes are Supplementary Table 1.

The density (number) of the insects used for diapause experiment must be indicated in the materials and methods!

All beet webworm species were kept in 28 cm × 18 cm × 13.5 cm boxes, 50 per box. This has been added to the revised manuscript.

Minor points

Line 165, the insects were placed in refrigerator at 4 °C for 3 weeks, which kept in dark or under light?

The insects were kept in total darkness when placed in a refrigerator at 4 ° C, which has been added in the revised manuscript.

Line 182, 2.2.1 Nucleic acid extraction and RNA-seq library construction No information of the library construction was provided in this section

Indeed, there is nothing about library construction in 2.2.1, just a description of the data used to construct the library, while the real library construction is the content of 2.2.2, including fragmented RNA, reverse transcription and cDNA synthesis, linking adaptor, etc. Therefore, "AND RNA-seq Library Construction" in 2.2.1 is deleted from the revised manuscript

Reference 4 and 41 are the same!

This error has been corrected in the new manuscript.

Round 2

Reviewer 1 Report (Previous Reviewer 2)

See attached pdf

Author Response

Thank you very much for your review of our manuscript, and I am very sorry for replying to you in such a format, cause your reply was marked on the original document.

At present, we have corrected the words, punctuation marks and other mistakes you mentioned in the article. Second, I checked where you marked the question mark and made changes.

For the Line68, 69 is not a sentence, we also revised it.

For the line 374 you annotated, the content annotated in line 386-388 should be the content for discussion, because it has been included in the discussion, we have deleted it.

We checked the article carefully and corrected other word errors, punctuation errors and so on.

Finally, thank you very much for your review of this manuscript.

Reviewer 2 Report (New Reviewer)

The Figures were disordered!

The references were not prepared according to the journal style!

The authors must check the spellings carefully through the whole manuscript before accepted for publication!

Line 17, delete the “HDAC” in the very end

Line 31 delete the “;” after the last key word

Line 45, How→how, after the semicolon, check the whole manuscript!

Line 49, remans → remains

Line 92 delete the “to diapause”

Line 95, delete the “induced into stasis”

Line 272, add “(Supplementary Table1)” after the “selected genes”

Line 275, comsidered→considered

Line 347 “but the decrease” →”but the increase”

Line 385, delete the comma between “makes” and “it difficult”

Line 456 “perdiapause” →”prediapause”

Line 470 “the author” →”We”

Line 486, add “.” after the “genes”

Line 502 “typiucally” →”typically”

Author Response

Thank you very much for taking time to work on this manuscript, and I am really sorry for the trouble to you. At present, we have revised this manuscript again, and we hope you can review it again.

The Figures were disordered!

The order of the Figures has been rearranged in the new manuscript.

The references were not prepared according to the journal style!

Since Endnote was linked before, the references automatically changed, so I modified the document format template of Endnote according to the journal style. The document format has been modified.

The authors must check the spellings carefully through the whole manuscript before accepted for publication!

In the new manuscript, the words have been spell checked and corrected.

Line 17, delete the “HDAC” in the very end

This has been deleted in the manuscript.

Line 31 delete the “;” after the last key word

This has been deleted in the manuscript.

Line 45, How→how, after the semicolon, check the whole manuscript!

This has been revised in the manuscript.

Line 49, remans → remains

This has been revised in the manuscript.

Line 92 delete the “to diapause”

This has been deleted in the manuscript.

Line 95, delete the “induced into stasis”

This has been deleted in the manuscript.

Line 272, add “(Supplementary Table1)” after the “selected genes”

This has been added in the manuscript.

Line 275, comsidered→considered

This has been revised in the manuscript.

Line 347 “but the decrease” →”but the increase”

This has been revised in the manuscript.

Line 385, delete the comma between “makes” and “it difficult”

This has been deleted in the manuscript.

Line 456 “perdiapause” →”prediapause”

This has been revised in the manuscript.

Line 470 “the author” →”We”

This has been revised in the manuscript.

Line 486, add “.” after the “genes”

This has been added in the manuscript.

Line 502 “typiucally” →”typically”

This has been revised in the manuscript.

We are very sorry that we made such a low-level mistake. We check the spellings carefully through the whole manuscript. And we have revised all the problems you mentioned above, and also revised the mistakes we found.

This manuscript is a resubmission of an earlier submission. The following is a list of the peer review reports and author responses from that submission.

Round 1

Reviewer 1 Report

The manuscript entitled “Transcriptome analysis of beet webworm reveals that histone deacetylase my affect diapause by regulating thermogenesis” by Cui Jin et al. represents a typical manuscript summarizing a wide differential transcriptomic screening. Typically, an exuberated number (>10,000) of differentially expressed genes is described. The authors tried to cluster genes according to GO and expression condition. Focus is directed to either light (photoperiod), temperature (chilling) or endocrine (JH1) regulated genes. Main outcome being the link between diapause and reduced metabolism. In accordance with similar manuscripts the overall info remains very general and superficial. Only the HDAC gene is named specifically.

Although photoperiod (short day condition) is known to be the main initiator of diapause the authors did not observe any effect upon light regulated genes which might indicate that the experimental setup was not optimal for studying this connection. Evidently larvae residing 5 cm underground will have minimal influence of changing light conditions.

In order of shedding some light upon diapause termination the authors included an experimental chilling condition resembling the temperature decrease during wintertime. Accordingly, they noticed a correlation with thermogenesis genes as well as with trehalose conversion (decrease in trehalose content post chilling and resulting in diapause termination. However, the authors do not explain the increase in trehalose during the course of diapause!

Detailed comments

1)      Supplementary data is incomplete! Only the same figures and their annotation is shown whereas tables explaining the true nature of the genes/gene sets discussed are missing!

2)      Abbreviations are not explained the first time cited e.g. WGCNA module line 275

3)      Species names must be written in italics all the time (lines 90, 353,357,401,406,407418,420,423438,443,452,456,469….)

4)      M&M and FIG 1 samples identification is very difficult to understand! Two different annotations are used (compare Fig1 and Fig 3!)

5)      Textual duplication hamper readability (see lines 185-186; lines 330-331)

6)      Fig 2 annotation is incomplete: what is meaning of 1,2,3,…8 in X-axis?

7)      Fig 3 each point represents mean of triplicate samples: No statistical variation???

8)      Fig3 JH1 is typically low during diapause but no steep increase at end of diapause is demonstrated? Is this really post diapause situation?

9)      Line 228 refers to Fig4 A which does not exist

10)   DEG total is 10,220 in line 212 and 11,486 in line 224?? Explain!

11)   Fig 5 I notice 4 sets of Venn diagrams What is the difference between each of them?  Authors mention” different stages” without any identification ??? Notice that numbers with the diagram are very small and difficult to read!

12)   Fig 6 not clear what to learn from these figures: missing supplementary data???

13)   Line 323-325? Therefore we believe HDAC may affect the expression of the histone deacetylase….. = Self induction???? What if “HDAC” should be replaced by either diapause or JH…..

14)   Fig7 not understandable if no explanation of the colors in the legend.

15)   Fig 9 both X – and Y-axis is named ordinate…

16)   Fig10 and fig11A are too small to read even after digital enlargement!

17)   Fig12 is unclear as info about the GO sets displayed is given!

18)   Lines 394-398 represent result interpretation and should be explained in the discussion section!

1)      Discussion is limited.

Reviewer 3 Report

The manuscript by Jin et al. addresses the transcriptional basis of diapause in the beet webworm Loxostege sticticalis, an important agricultural pest.  The study describes unique transcriptional patterns across five developmental stages: pre-diapause, early diapause, late diapause, late diapause with a temperature chill and post-diapause.  The data presented in the manuscript has the potential to make a contribution to the research literature on the transcriptional basis of diapause. However, the writing is hard to follow, has multiple typographic errors, and lacks clarity and critical detail in regards to how the experiments were conducted.  These issues make it impossible to assess the results and evaluate whether the conclusions are supported.  Overall, the manuscript requires extensive re-organization and re-writing before it can be re-evaluated.       

  • The introduction does not clearly define specific question(s) and how they were tested.  For example the authors mention that the role of temperature throughout diapause in the beet webworm is unclear, but they do not explain how temperature is incorporated to test a specific question.  Similarly, the authors do not explain how JH, FA and Trehalose will be used in the study design.  

  • A major concern is the lack of attention to detail when describing the materials and methods.  For example there is no indication of sample size, sample preparation and replication for most methods.  The authors need to explain what constitutes a sample (i.e. single whole individuals vs pooled individuals), how samples were prepared, and how many biological replicates were used per treatment for each analysis.  

  • The conditions used for diapause induction lack precision and are unclear and counterintuitive.  Diapause induction is described as occurring under light conditions of L:D =16:8 (long-day conditions) from the first to fourth instar.  Light conditions are then switched to L:D=12:12 after the end of the fourth instar, when the webworm should already be in diapause.  The authors need to clarify which is the photosensitive stage and when diapause begins (the introduction only states webworms enter diapause as “late mature larvae”).  

  • It is unclear how treatment of the control group differed from the experimental groups throughout the rearing process and there is no mention of how the control group was used in any downstream analysis.

  • Additional crucial details are missing from both the methods and results.  There is no description of how sequencing libraries were constructed, what instrument was used for sequencing, how many reads were obtained, what pipeline was used for read mapping, nor what percentage of reads mapped to the new assembly.  Further, in addition to describing the use of software in sections 2.6-2.9, the authors need to describe the steps of their analyses and parameters in more detail.  For example, in section 2.7 there is no explanation of how modules were analyzed “in relation” to trait information, and in section 2.9 there is no description of control primers used for normalization, nor an explanation of the method used for qPCR quantification. 

  • Figures require extensive editing.  For example the acronyms in Figure 1 are not intuitive and are not defined in the caption.  Further, it is not clear what the tick marks between “NBS” to “D_Mail1” and “Chil” to “Rel_D” refer to, and it is not clear from the figure which stages correspond to diapause (as depicted diapause is only indicated under “D_Mail1”, though presumably diapause should persist through the “Chil” stage).  Additionally, the placement of “Low temperature at 0oC” is unclear.  Are only animals collected for the “Chil” treatment exposed to this temperature or do all animals collected after D_Mail2 experience this temperature?  Further, the temperature used at other stages is not depicted.  If the figure is intended to describe the experimental design, the caption also needs to concisely address how animals were reared and sampled for each stage, and should clarify the photoperiod for all stages.  It would also be crucial to depict the photosensitive stage/treatment in the figure as that is critical to the design. 
    • All other figures are misnumbered in-text and are often lacking axis titles and descriptive legends.  For example, Figure 2 is referenced in relation to text about numbers of differentially expressed genes between stages, but the figure caption describes it as results of RT-PCR analysis.  In any case, it has no label for either axis and no indication of which transcripts were being measured, nor which genes were used as references for normalization.  Presumably all treatments were compared relative to NBS but that needs to be explicitly stated.  Further, there is no indication of variance in the data and no evaluation of statistical significance.  All figures must be edited to ensure axes are labeled, variance is depicted, captions are descriptive and images are readable (note, for example, that boxes in Figure 10 are entirely unreadable and the purpose of the graphic is not clear). 

  • Given the lack of detail and clarity in the methods and results, the conclusions are not possible to evaluate.  In any case, a discussion of the results should begin by reiterating the focus of the study and experiments, and should address the main transcriptional and physiological results before presenting supporting examples from other species.   

Round 2

Reviewer 1 Report

The authors improved the manuscript by correct and extensive reacting to the reviewer comments. 

As well intro, M&m, fig annotations, discussion, references are elaborated in accord to comments.

Supplementary data are now correctly displayed

Reviewer 2 Report

The paper needs to be completely re-organised and graphical presentations need to be reassessed.  Figure 10 is completely unreadable and the importance of the various figures are not clearly stated.  The reader is simply having data dumped into the paper without any indication of relative importance.  There is still the difficulty with understanding how the genes that correlate with JH are important as no change except an initial decrease is shown in JH, so correlations are meaningless.

Further re-working of the manuscript is necessary before it meets the requirement of being publishable.  

Also, please check your reference 2, as no insect is called Occsta migratoria, but locusts are Locusta migratoria.  This alone suggests that you had not checked your references adequately.

Reviewer 3 Report

Although the authors have made some revisions to this manuscript, it remains unpublishable.

In the revised manuscript additional information is provided re: the samples used for the RNAseq study, but no information is provided regarding the samples used for the “relevant physiological indicators” (free fatty acids, trehalose and JH) i.e., how many animals per biological replicate? how many biological replicates?

Simalarily, the quantitative reverse PCR experiments, no information is provided regarding the biological samples used- i.e., how many animals per biological replicate?  How many biological replicates?

Figures 10, 11, and 12remain impossible to read and interpret.

My main suggestion to the authors it to find a colleague with experience publishing papers in English language journals, and work with them to carefully revise the manuscript so that the methods and results are clearly presented.  If the methods and results can not be easily understood, then it is impossible to determine if the conclusions of the study are justified.

Under these conditions I am unwilling to provide a detailed review of this manuscript and will not review future versions.